# Spontaneous Rupture of Esophageal Diverticulum—A Case Report and Literature Review

**DOI:** 10.3390/diagnostics13010019

**Published:** 2022-12-21

**Authors:** Shengting Dong, Shiliang Xie, Yongxin Zhou

**Affiliations:** Department of Thoracic-Cardiovascular Surgery, Tongji Hospital, School of Medicine, Tongji University, 389 Xincun Rd, Shanghai 200065, China

**Keywords:** esophageal, oesophageal, spontaneous esophageal rupture, spontaneous esophageal diverticulum rupture

## Abstract

The spontaneous rupture of the esophageal diverticulum is a rare condition that occurs without any warning signs. Its incidence is low, but the mortality rate is high. This paper reports a case of spontaneous esophageal diverticulum rupture and analyzes it along with 13 other cases to explore its prevention and treatment measures. When patients suffer from chronic swallowing difficulties and chest pain or vomiting that cannot be explained after meals, they should be suspected to have a possible spontaneous rupture of the esophageal diverticulum, which is critical to the patient’s prognosis.

Esophageal diverticulum is a relatively rare benign condition of the esophagus, and the presence of an esophageal diverticulum is frequently accompanied with esophageal symptoms such as dysphagia and regurgitation. Spontaneous rupture of the esophageal diverticulum is a non-traumatic rupture of the full floor of the esophageal diverticulum wall and it is mostly caused by a sudden increase in esophageal pressure [1]. Because the esophagus is close to the lungs and heart, the rupture of the esophageal diverticulum is usually dangerous. However, there is no standardized treatment for spontaneous rupture of the esophageal diverticulum.

Therefore, this article reports a case of spontaneous esophageal diverticulum rupture in our hospital, and summarizes the diagnosis and treatment methods of spontaneous esophageal diverticulum rupture on the basis of literature analysis.

This case study focuses on a female patient who is 74 years old. On 25 June 2020, she came to see a doctor for sudden upper abdominal pain and back pain after a meal. Her computer tomography (CT) (Figure 1) showed that there was a liquid pneumothorax in the right pleural cavity, with free gas in the mediastinum and neck, and scattered inflammation in both lungs. Therefore, she was given closed chest drainage, anti-infective and supportive care. Since the chest drain contained food debris, she was given a gastric tube to relieve pressure. On 28 June 2020, after her condition stabilized, the esophagography (Figure 2) showed a bag -shaped change on the right side of the chest, and there was a relative barium meal leakage. On 29 June 2020, her upper gastrointestinal endoscopy (Figure 3a,b) showed that there was an esophageal diverticulum opening in the esophageal wall 35 cm from the incisors. Considering the patient’s age and diverticulum rupture time and no other risk factors identified, she was given conservatively continued treatment with enteral nutrition and gastric decompression drainage with a three-lumen feeding tube. The drainage effect was good, and the inflammation was significantly reduced. On 31 July 2020, her upper gastrointestinal endoscopy (Figure 4) showed that the opening of the diverticulum had basically closed. After that, the patient had no particular discomfort, and her condition gradually stabilized. She was discharged from the hospital and asked to return for re-examination after half a month. Two years later, we called to enquire about her situation and found that she was still in good health.

The Web of Science database and PubMed database were searched for articles from 1971 to 2022. Ultimately, 13 cases met our inclusion criteria. Finally, the reported general demographic data, clinical presentation, diagnosis, treatment options, and prognosis were collected, recorded, and summarized (Table 1 and Table 2). We examined the relevant literature and outlined a few crucial problems that clinicians should be aware of.

Esophageal diverticulum is very rare, with an incidence rate of less than 1% of the total population [15]. As show in Table 1 and Table 2, a total of 13 patients with spontaneous esophageal diverticulum rupture were reported in 13 studies and, including our case, a total of 14 patients with an average age of 61.7 years (50–76 years), including 12 (85.7%) male patients and 2 (14.3%) female patients, and most cases (64.3%) had no obvious cause. Studies have shown that the cause of esophageal rupture is mainly related to severe vomiting after meals [16] and, in our study, only one case was a diverticulum rupture after vomiting. In addition to this, including our case, there were six patients with diverticulum rupture associated with meals.

As show in Table 2, chest pain (42.9%), progressive dysphagia (42.9%), and epigastric pain (35.7%) were the main symptoms of most patients. Although vomiting, chest pain, and subcutaneous emphysema are the Mackler triad of esophageal rupture [17], in our study, only one patient had vomiting symptoms. In addition, our study found that these 14 patients had a relatively high confirmed diagnosis rate (78.6%) during the diagnosis process. For the treatment of spontaneous esophageal diverticulum rupture, surgery was performed in the majority of cases (71.4%). In many studies, the mortality rate of esophageal rupture is high, above 20% [18,19,20]. However, the mortality rate of esophageal diverticulum rupture was only 7.1% in our study. This may be related to the high confirmed diagnosis rate in our study and the individualized treatment approach used in all cases. The above findings show that there might be new inspiration for the treatment of esophageal diverticulum rupture.

Studies have shown that supraphrenic diverticulum mostly occurs on the right side of the esophagus [21,22], and our research has also confirmed this. In our study, esophageal diverticula were mostly located in the lower segments (50%) and right side of the esophagus (57.1%). Many studies have shown that spontaneous esophageal rupture usually occurs in the lower third of the esophagus and on the left side of the esophagus, which is related to the thin muscle layer of the left side of the esophagus [23,24]. However, as shown in Table 1, in our study, half of the spontaneous esophageal diverticulum ruptures on the right side of the esophagus are located in the lower segments of the esophagus. Moreover, studies have shown that epiphrenic diverticula located in the lower esophagus are mostly located on the right side of the esophagus [22]. As shown in this study, the location of the esophageal diverticulum is related to the adjacent mediastinal structures. For the diagnosis of esophageal diverticulum ruptures, chest X-ray and CT can only diagnose perforation but not confirm diagnosis, and esophagography and upper gastrointestinal endoscopy are the most accurate ways of diagnosis [25]. In our study, most cases were diagnosed by esophagography (71.4%) and upper gastrointestinal endoscopy (57.1%).

The treatment of esophageal diverticulum rupture is often formulated according to the treatment plan and specific conditions of the esophageal diverticulum.

Considering that most of the patients are elderly, non-surgical conservative treatment is the first choice when surgery cannot be tolerated or the optimal time for surgery is missed. Non-surgical conservative treatment is mainly catheter drainage, anti-infection and supportive care. The catheter mainly includes a chest drainage tube, gastric tube, and three-lumen feeding tube. Their main role is to reduce gastric acid irritation, relieve pressure in the esophagus, and provide enteral nutrition. According to our experience, if conditions permit, a gastric tube is placed in the esophageal diverticulum to drain, and the treatment effect will be better. Drugs mainly include acid suppressants and anti-inflammatory drugs. Our patient was treated with non-surgical conservative treatment because of her old age. In addition, her esophageal diverticulum has been ruptured for a long timeand no risk factors for gastric tube drainage in the diverticulum., Therefore, she recovered well.

Spontaneous esophageal diverticulum rupture is rare. It occurs mostly in senior men and ruptures of the esophageal diverticulum in the right lower esophagus are the most common. At present, there is no standardized treatment for esophageal diverticulum rupture. The treatment method is mainly based on the treatment of esophageal diverticulum and individualized treatment methods are formulated according to the situation of the patient’s diverticulum rupture. Furthermore, there are several key points that physicians should keep in mind when caring for a patient with spontaneous esophageal diverticulum rupture. First, spontaneous rupture of esophageal diverticulum is mostly associated with chronic swallowing difficulties and chest pain or vomiting after meals that cannot be explained. Second, when patients cannot tolerate surgery or miss the optimal time for surgery, we may consider non-surgical treatment, in which infection control is very important. Finally, regardless of surgical or non-surgical treatment, early confirmed diagnosis and individualized treatment are crucial for favorable prognosis.

## Figures and Tables

**Figure 1 diagnostics-13-00019-f001:**
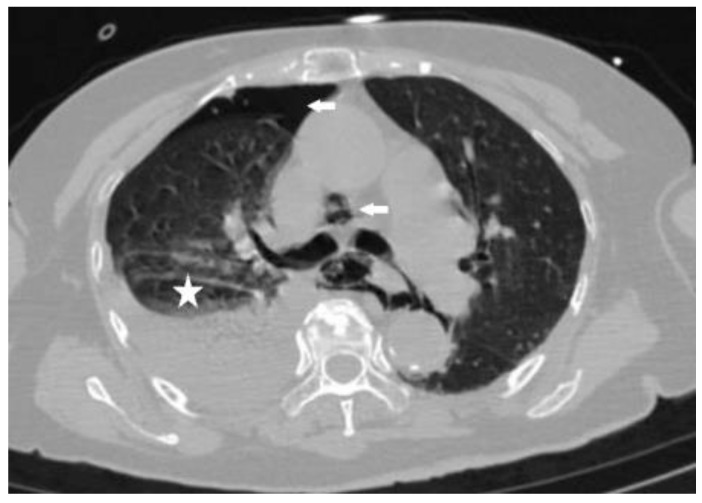
Computed tomography (CT) of the chest shows the gas in the right thoracic cavity and mediastinum (arrows) and right pleural effusion (star).

**Figure 2 diagnostics-13-00019-f002:**
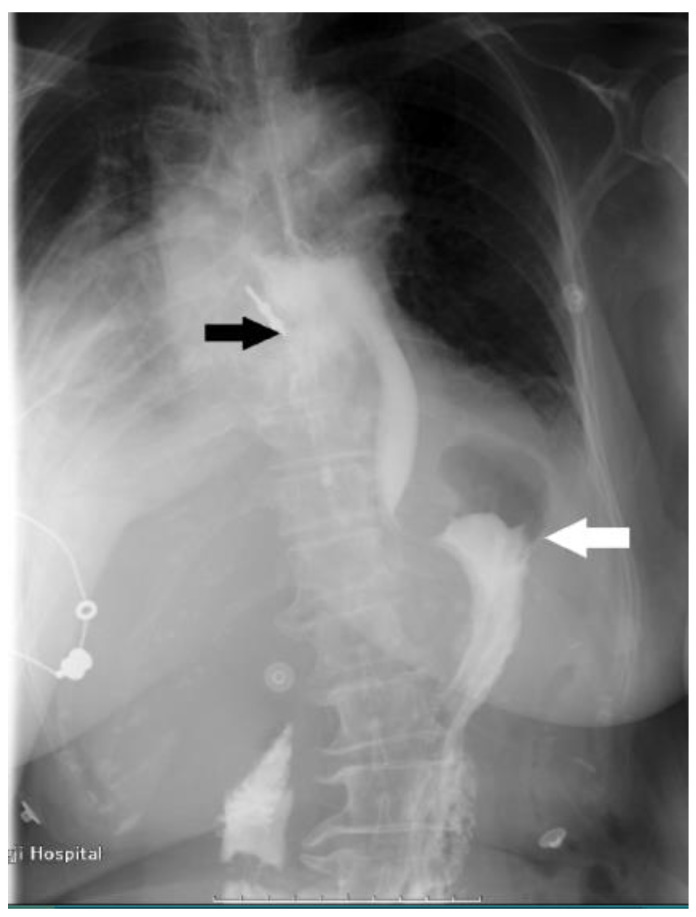
The esophagography shows leakage of contrast medium (white arrow), barium meal leakage, and esophageal diverticulum on the right side of the esophagus (black arrow).

**Figure 3 diagnostics-13-00019-f003:**
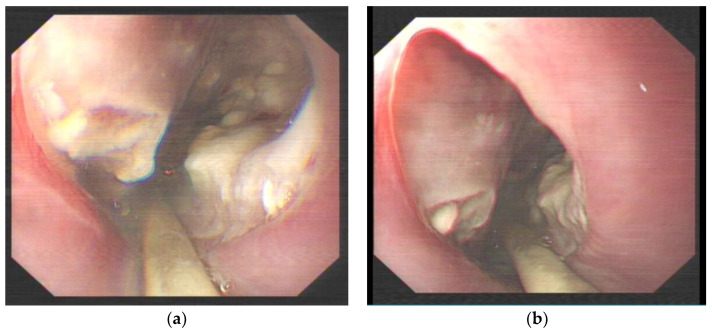
(**a**) Upper gastrointestinal endoscopy showing the opening of the esophageal diverticulum 35 cm from the incisors, in which a large amount of necrosis was observed. (**b**) Upper gastrointestinal endoscopy showing a smooth esophageal wall connected to the esophageal diverticulum.

**Figure 4 diagnostics-13-00019-f004:**
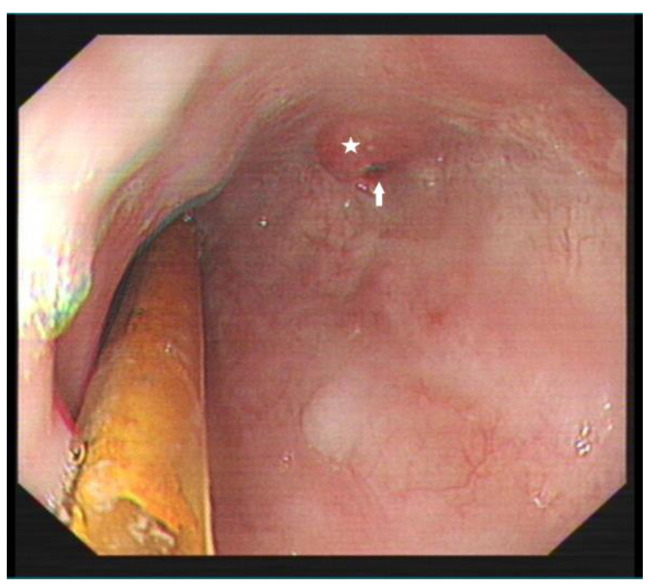
Re-examination of upper gastrointestinal endoscopy showing an esophageal diverticulum opening measuring 0.5 × 0.3 cm (arrow) with surrounding scar-like hyperplasia (star).

**Table 1 diagnostics-13-00019-t001:** Baseline characteristics of 13 patients with spontaneous rupture of esophageal diverticulum.

Year	Author	Age (Years)	Sex	Location		Diagnostic or Not	Treatment	Outcome
1985	Gutiérrez del Olmo et al. [2]	74	F	ME	R	N	NST	Cured
1996	Oka et al. [3]	53	M	ME	R	Y	NST and then ST	Cured
2000	Pistorius et al. [4]	76	M	ME	L	Y	ST	Died
2000	Murakami et al. [5]	61	M	LE	A	Y	ST	Cured
2003	Cantù. [6]	75	M	LE	R	Y	ST	Cured
2004	Lee et al. [7]	68	M	LE	L	Y	ST	Cured
2006	Mecklenburg et al. [8]	40	M	ME	Unclear	Y	Self-healing	Cured
2009	Hung. [9]	50	M	UE	R	N	NST	Unclear
2014	Almre et al. [10]	53	M	UE	P	Y	ST	Cured
2018	Onodera et al. [11]	58	M	LE	R	Y	ST	Cured
2019	Ding et al. [12]	64	M	LE	L	N	ST	Cured
2021	Morita et al. [13]	51	M	UE	L	Y	ST	Cured
2022	Wangkulangkul et al. [14]	67	M	LE	R	Y	ST	Cured

F: Female, M: Male, UE: Upper segments of esophagus, ME: Middle segments of esophagus, LE: Lower segments of esophagus, R: Right side of esophagus, L: Left side of esophagus, A: Anterior side of esophagus, P: Posterior side of esophagus, N: No, Y: Yes, NST: Non-surgical treatment, ST: Surgical treatment.

**Table 2 diagnostics-13-00019-t002:** Summary features of 14 patients with spontaneous rupture of esophageal diverticulum.

Characteristic	Value
**Age, year**	61.7 (50–76)
**Sex**	
Male	12 (85.7%)
Female	2 (14.3%)
**Location**	
Upper segments of esophagus	3 (21.4%)
Middle segments of esophagus	4 (28.6%)
Lower segments of esophagus	7 (50%)
Left side of esophagus	4 (28.6%)
Right side of esophagus	8 (57.1%)
Anterior side of esophagus	1 (7.1%)
Posterior side of esophagus	1 (7.1%)
**Etiology**	1 case was caused by vomiting (7.1%) 1 case was caused by eosinophilic esophagitis (7.1%)
**Treatment**	
Non-surgical treatment	4 (28.6%)
Surgical treatment	10 (71.4%)
Esophageal diverticulum repair	5 (35.7%)
Esophagectomy	4 (28.6%)
Magnet-assisted endoscopic diverticulotomy	1 (7.1%)
**Misdiagnosis**	3 (21.4%)
**Confirmed**	11 (78.6%)
**Outcome**	
Cured	12 (85.7%)
Died	1 (7.1%)
Unclear	1 (7.1%)

## Data Availability

Data available on request due to privacy restrictions.

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
