# Peer review of "Spontaneous Rupture of Esophageal Diverticulum—A Case Report and Literature Review"

_diagnostics, 2022, doi:10.3390/diagnostics13010019_

Round 1
Reviewer 1 Report
Thank you for the interesting case report and the endoscopic images, but there have been many reports on esophageal diverticula and esophageal diverticulum rupture so far, and it is considered to be of little novelty.
The authors note in Discussions 109-114 and 138-142 that the long-term presence of diverticula results in the formation of supporting tissue.
In addition, it states that conservative treatment is an option instead of surgery for that purpose, but I think this is an exaggeration, because it does not present the condition around the diverticulum in this case, and it is the result of only a small number of cited cases.
Author Response
On behalf of all the contributing authors, I would like to express our sincere appreciations of your letter and reviewers 'constructive comments concerning our manuscript entitled “Spontaneous Rupture of Esophageal Diverticulum—a Case Report and Literature Review "(ID: diagnostics-2022262). Those comments are all valuable and very helpful for revising and improving our paper, as well as the important guiding significance to our researches. According to the associate editor and reviewers' comments, we have made extensive modifications to our manuscript and supplemented extra data to make our results convincing. In this revised version, changes to our manuscript were all highlighted within the document by using red-colored text.
We would like to re-submit this revised manuscript to diagnostics, and we hope it is acceptable for publication in the journal.
We are looking forward to hearing from you soon.
1. Response to comment: (There have been many reports on esophageal diverticula and esophageal diverticulum rupture so far, and it is considered to be of little novelty.) Response: Thank you for your comments on our study. Although there are many related articles, no paper has so far conducted a comprehensive review of the literature on spontaneous esophageal diverticulum rupture. The objective of this article is to discuss the clinical characteristics and identification points of spontaneous rupture of esophageal diverticulum, and discuss its prevention and treatment measures in combination with cases. Therefore, we hope that you can give us some suggestions on the basis of our revised manuscript. Thank you.
2. Response to comment: (The authors note that the long-term presence of diverticula results in the formation of supporting tissue. It states that conservative treatment is an option instead of surgery for that purpose, but I think this is an exaggeration.)
Response: Thanks for your constructive suggestions. We also feel that our presentation is not rigorous enough, so we have adjusted the relevant content. In addition, for the treatment of spontaneous esophageal diverticulum rupture, we did not replace surgical treatment with conservative treatment in the article. This may be a problem caused by unclear English language during the editing process of our manuscript. We also hope that you can give us your valuable advice again. (Page 6, Lines 130-135; Page7, Lines 164-176).

Reviewer 2 Report
1. Overview of different types pf esophageal diverticula would enhance the article.
2. Case description (line 26-43) is presented in poor language.
3. Table 2: Consider capital letters for "sex" and "esophagectomy"
Author Response
On behalf of all the contributing authors, I would like to express our sincere appreciations of your letter and reviewers 'constructive comments concerning our manuscript entitled “Spontaneous Rupture of Esophageal Diverticulum—a Case Report and Literature Review "(ID: diagnostics-2022262). Those comments are all valuable and very helpful for revising and improving our paper, as well as the important guiding significance to our researches. According to the associate editor and reviewers' comments, we have made extensive modifications to our manuscript and supplemented extra data to make our results convincing. In this revised version, changes to our manuscript were all highlighted within the document by using red-colored text.
We would like to re-submit this revised manuscript to diagnostics, and we hope it is acceptable for publication in the journal.
We are looking forward to hearing from you soon.
1. Response to comment: (Overview of different types pf esophageal diverticula would enhance the article.)
Response: Thank you for your comments on our study. We had the same idea as you in the early stage of the study, but we did not outline different types of esophageal diverticulum in the manuscript for the following reasons: first, only 14 cases were consistent with our article, and the number of relevant cases was relatively small, so the separate analysis was not rigorous and reasonable. Second, the correlation between the same types of esophageal diverticula is low, and no reasonable conclusions can be drawn. We hope that you can give us some suggestions on the basis of our revised manuscript.
2. Response to comment: (Case description (line 26-43) is presented in poor language.)
Response: Thanks for your constructive suggestions. We have consulted the relevant description and revised the manuscript. (Page 1, Lines 30-46; Page 1, Lines 47-49).
3. Response to comment: (Table 2: Consider capital letters for "sex" and "esophagectomy".)
Response: Thanks for your constructive suggestions. We have corrected the writing error. Finally, we also hope that you can give us some suggestions on the basis of our revised manuscript. (Table 2).

Reviewer 3 Report
The case report with the literature review is presented.
The additional check of English language is desirable. For example it seems that in the sentence "However, there is no standardized treatment for spontaneous esophageal diverticulum." the word "rapture" is missed.
The article may be interesting to the surgeons, practicing in emergency.
Author Response
On behalf of all the contributing authors, I would like to express our sincere appreciations of your letter and reviewers 'constructive comments concerning our manuscript entitled “Spontaneous Rupture of Esophageal Diverticulum—a Case Report and Literature Review "(ID: diagnostics-2022262). Those comments are all valuable and very helpful for revising and improving our paper, as well as the important guiding significance to our researches. According to the associate editor and reviewers' comments, we have made extensive modifications to our manuscript and supplemented extra data to make our results convincing. In this revised version, changes to our manuscript were all highlighted within the document by using red-colored text.
We would like to re-submit this revised manuscript to diagnostics, and we hope it is acceptable for publication in the journal.
We are looking forward to hearing from you soon.
1. Response to comment: (The case report with the literature review is presented and the article may be interesting to the surgeons, practicing in emergency.)
Response: Thank you for your approval of my article.
2. Response to comment: (The additional check of English language is desirable. For example it seems that in the sentence "However, there is no standardized treatment for spontaneous esophageal diverticulum.)
Response: Thanks for your constructive suggestions. We have corrected the writing error. Finally, we also hope that you can give us some suggestions on the basis of our revised manuscript. (Page 1, Lines 25).

Round 2
Reviewer 1 Report
I think that appropriate corrections have been made to the treatment strategy, which I was most concerned about.
Depending on the patient's condition, it is reasonable that conservative treatment may be an option in difficult-to-operate cases.
Thank you for your correction.
The authors state that rupture often occurs on the right wall. Is it possible to consider this point?
In addition, Boerhaave syndrome is said to be more common on the left side of the esophagus. It has been pointed out that the reason for this is that the muscle layer on the left side is thin. Can you compare and consider it?
Also, I think that there are diverticula with and without muscle layers. Is it possible to consider this? If it is possible to consider these points, I think it would be better to discuss the clinical characteristics and identification points of spontaneous rupture of esophageal diverticulum.
Author Response
1. Response to comment: (The authors state that rupture often occurs on the right wall. Is it possible to consider this point?)
Response: Thank you for your comments on our study. We think it is possible to consider this point, and the above references also mention that spontaneous esophageal diverticulum rupture mostly occurs on the right side of the esophagus. This is also consistent with the supplementary literature that we found at a later stage. Therefore, we hope that you can give us some suggestions on the basis of our revised manuscript. Thank you. (Page 6, Lines 141-143; Page7, Lines 144-149).
2. Response to comment: (Boerhaave syndrome is said to be more common on the left side of the esophagus.It has been pointed out that the reason for this is that the muscle layer on the left side is thin.Can you compare and consider it?)
Response: Thanks for your constructive suggestions. We found that spontaneous esophageal rupture was indeed more common on the left side of the esophagus because of the thin muscle layer on the left side. However, our research and literature review have found that there are few reasons for spontaneous esophageal diverticulum rupture on the right side of the esophagus, so the comparison between them can only be limited to the causes that have been reported in the literature. We also hope that you can give us your valuable advice again. (Page 6, Lines 141-143; Page7, Lines 144-149).
3. Response to comment: (I think that there are diverticula with and without muscle lavers. Is it possible to considerthis?)
Response: Thanks for your constructive suggestions. This was noted earlier in our study, but the literature we searched did not describe whether the esophageal diverticulum had a muscular layer. In addition, we found that the location of the esophageal diverticulum was largely related to its adjacent mediastinal structure. Because the main purpose of our article is to analyze the clinical characteristics of spontaneous esophageal diverticulum rupture and give guiding treatment recommendations around the cases we have found, we can only describe its location in a few paragraphs. We also hope that you can give us some suggestions on the basis of our revised manuscript. Thank you. (Page 6, Lines 141-143; Page7, Lines 144-149).
